# Stem cell progeny contribute to the schistosome host-parasite interface

**James J Collins III[1,2,3]\*, George R Wendt[2], Harini Iyer[1,3], Phillip A Newmark[1,3]\***

[1]Department of Cell and Developmental Biology, University of Illinois at Urbana-Champaign, Urbana, United States; [2]Department of Pharmacology, UT Southwestern Medical Center, Dallas, United States; [3]Howard Hughes Medical Institute, University of Illinois at Urbana-Champaign, Urbana, United States

**Abstract** Schistosomes infect more than 200 million of the world's poorest people. These parasites live in the vasculature, producing eggs that spur a variety of chronic, potentially life-threatening, pathologies exacerbated by the long lifespan of schistosomes, that can thrive in the host for decades. How schistosomes maintain their longevity in this immunologically hostile environment is unknown. Here, we demonstrate that somatic stem cells in *Schistosoma mansoni* are biased towards generating a population of cells expressing factors associated exclusively with the schistosome host-parasite interface, a structure called the tegument. We show cells expressing these tegumental factors are short-lived and rapidly turned over. We suggest that stem cell-driven renewal of this tegumental lineage represents an important strategy for parasite survival in the context of the host vasculature.

**\*For correspondence:** JamesJ. Collins@UTSouthwestern.edu (JJC); pnewmark@life.illinois.edu (PAN)

**Competing interests:** The authors declare that no competing interests exist.

## Introduction

Neoblasts are pluripotent stem cells essential for regeneration and tissue homeostasis in a variety of free-living flatworms, most notably freshwater planarians (*Newmark and Sánchez Alvarado, 2002*; *Wagner et al., 2011*). Previously, it was shown that schistosomes, like their free-living relatives, also possess neoblasts, capable of self-renewal and differentiation into tissues such as the intestine and muscle (*Collins et al., 2013*). However, the role these cells play in the biology of the parasites in their mammalian host was unexplored. To decipher the cellular functions of schistosome neoblasts, we compared the short-term and long-term transcriptional consequences for the parasite following neoblast depletion (*Figure 1a*).

## Results and discussion

To examine the transcriptional effects of neoblast ablation, we exploited the observation that expression of genes in differentiated tissues (e.g., the intestine) is unaffected at 48 hr following irradiation, whereas the neoblasts are irreversibly killed (*Collins et al., 2013*). Previously, we demonstrated that many genes down-regulated at 48 hr following irradiation were associated with the schistosome neoblasts (*Collins et al., 2013*). Thus, we reasoned that by comparing the gene expression profiles of parasites shortly after neoblast ablation (48 hr) to parasites two weeks after their neoblasts had been killed, we could characterize the long-term consequences of neoblast depletion. Specifically, we expected genes down-regulated at both early and late time points to be neoblast-enriched factors, whereas genes only down-regulated at later time points would be genes that require neoblasts for maintaining their expression. To add specificity to this dataset, removing genes whose expression could be influenced non-specifically by irradiation (*Solana et al., 2012*), we also profiled parasite transcriptomes after long-term RNA interference (RNAi) targeting either of two

**eLife digest** Schistosomes are parasitic worms that infect and cause chronic disease in hundreds of millions of people in the developing world. A major reason these parasites are so damaging is that they are capable of living and reproducing in the human bloodstream for decades.

Previous research had shown that schistosomes have a population of stem cells that are proposed to promote the parasite's survival inside the host's bloodstream. However, it was not clear what role these cells play in the worms.

Collins et al. have now found that, in a parasitic worm called *Schistosoma mansoni*, a large number of these stem cells are destined to become cells that generate the parasite's skin. This unique tissue is known as the tegument, and had long been thought to have evolved in parasitic flatworms to help them survive in their host and evade its immune defenses. Therefore, Collins et al.'s findings suggest a new mechanism by which stem cells can promote the survival of a parasite inside its host.

In the long-term, these findings could lead to new treatments for parasitic infections and may shed light on the evolution of parasitic flatworms. An important future challenge will be to determine if disrupting these parasites' stem cells, and their ability to generate new tegumental cells, has any effect on the parasite inside its host.

genes required for the maintenance of proliferating neoblasts: *fgfrA* (*Collins et al., 2013*) or *histone H2B* (*Figure 1—figure supplement 1*).

From our transcriptional profiling experiments of male somatic tissues we identified 135 genes that were down regulated ($\geq$1.25x, p<0.05) in both our irradiation and RNAi datasets (*Figure 1b*, *Supplementary file 1*). As anticipated, this gene set included a number of known stem cell- (e.g., *nanos2* and *ago2*-1[*Collins et al., 2013*]) and cell cycle-specific (*cyclinB* and *mcm2*) genes (*Figure 1b,c* and *Supplementary file 1*). More importantly, we identified 105 genes that were not down-regulated at D2 post-irradiation but were significantly down regulated ($\geq$1.25x, p<0.05) at D14 post-irradiation and following RNAi of either *fgfrA* or *histone H2B* (*Figure 1b,c* and *Supplementary file 1*). For brevity, we will refer to these 105 genes as delayed irradiation-sensitivity (DIS) genes. We also noted a small class of genes that were modestly down regulated at early time points and highly down regulated after long-term stem cell depletion (*Supplementary file 1*). The most striking example of this class was the schistosome orthologue of the planarian *p53* (*Pearson and Sánchez Alvarado, 2010*), which was down-regulated ~2 fold at 48 hr and nearly 150 fold at D14 post-irradiation (*Supplementary file 1*).

To validate our transcriptional profiling experiments, we examined a subset of these DIS genes by whole-mount in situ hybridization at D2 and D7 following irradiation. As anticipated, expression of *cathepsin B,* a gene expressed in differentiated intestinal cells, was unaffected at either time point (*Figure 2a*). Conversely, the expression of genes associated with the neoblasts (*fgfrA* and *nanos2*) was substantially reduced at D2 and the expression of these genes did not return by D7 (*Figure 2a*), confirming that stem cells are irreversibly depleted by irradiation. Consistent with our RNAseq data, the number of cells expressing *p53* is modestly reduced at D2 post-irradiation and dramatically reduced by D7 (*Figure 2a*). In contrast to the neoblast-expressed genes and *p53*, at D2 the number of cells expressing the DIS genes *tsp-2, sm13, sm29,* and *val-8* was unaffected (*Figure 2a*). However, by D7 post-irradiation the expression of these genes was severely depleted (*Figure 2a*). We did note in our RNAseq experiments, and in independent qPCR experiments, a modest increase in *val-8* mRNA levels 48 hr post-irradiation (*Figure 1* and *Figure 1—figure supplement 2*). Since, the number of *val-8*[+] cells did not appear to dramatically change at 48 hr post-irradiation (*Figure 2a*), it is possible that some *val-8*[+] cells had elevated levels of the *val-8* mRNA. To directly examine the relationship between genes expressed in neoblasts and the DIS genes, we performed double fluorescence in situ hybridization (FISH) experiments with *histone H2B, p53,* and *tsp-2*. We observed no co-expression of the DIS gene *tsp-2* with *histone H2B* *Figure 2—figure supplement 1*, suggesting that DIS genes are expressed in a population of cells other than neoblasts. Consistent with our observations following irradiation, we observed that *p53* was expressed in both the *histone H2B*[+] neoblasts

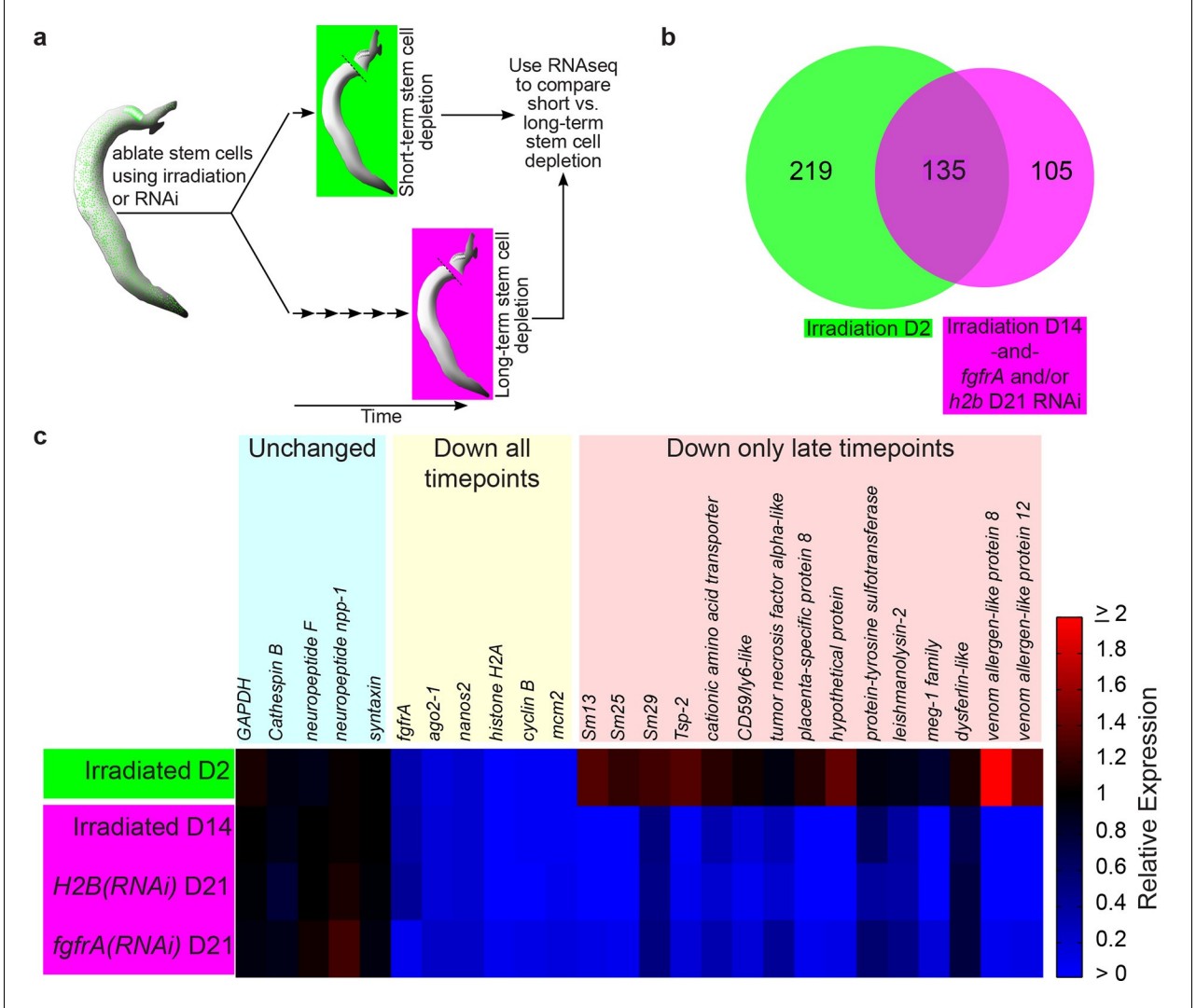

**Figure 1.** Identification of genes down-regulated after long-term stem cell depletion. (a) Scheme for transcriptional profiling studies. (b) Venn Diagram showing number of genes significantly down-regulated after short-term (green) and long-term (magenta) stem cell depletion. (c) Heat map showing relative gene expression for various treatments and time points. Only a subset of representative genes is displayed.

The following figure supplements are available for figure 1:

**Figure supplement 1.** *histone H2B* is required to maintain proliferative neoblasts.

**Figure supplement 2.** *val-8* expression is increased 48 hr following irradiation.

and *tsp-2*[+] cells (*Figure 2—figure supplement 1*). Together, these data strongly support the model that the DIS genes are expressed in an irradiation-sensitive population of cells that is molecularly distinct from the neoblasts.

Upon closer examination we noted that a number of the DIS genes encoded proteins previously shown by immunological and/or proteomic approaches to be associated with the parasite's surface (e.g., *tsp-2* (*Tran et al., 2006*; *Pearson et al., 2012*; *Wilson, 2012*), *sm13* (*Abath et al., 2000*; *Wilson, 2012*), *sm29* (*Braschi and Wilson, 2006*; *Cardoso et al., 2008*; *Wilson, 2012*), *sm25* (*Abath et al., 1999*; *Castro-Borges et al., 2011*; *Wilson, 2012*)). The schistosome surface is covered by a continuous syncytial structure, called the tegument (*Figure 2b*), which serves as the primary barrier between the parasite and its host. This unique tissue is connected by cytoplasmic bridges to

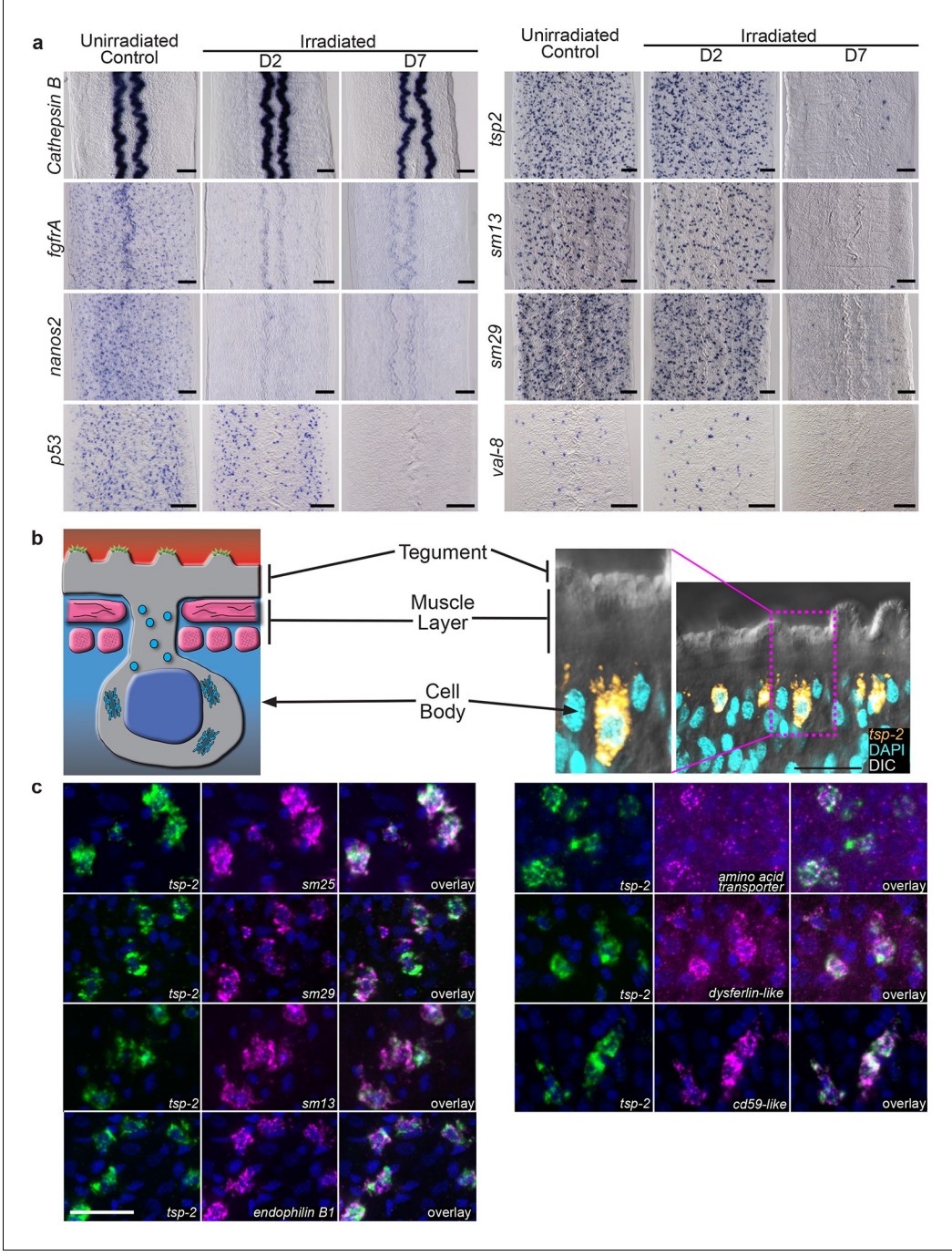

**Figure 2.** Cells expressing DIS genes are lost following stem cell depletion and express genes associated with the schistosome tegument. (a) Whole-mount in situ hybridization to detect genes expressed in: the intestine (*Cathepsin B*); neoblasts (*fgfrA*, *nanos2*); or cells expressing DIS genes (*tsp-2*, *sm13*, *sm29*, *val-8*) in either untreated parasites or worms at D2 or D7 following irradiation. *p53* is also shown as an example of a gene modestly down-regulated at early time points and highly down-regulated at late time points after neoblast ablation. Expression of DIS genes is unaffected at day 2 following irradiation but is substantially reduced by day 7. n > 3 for each data point. (b) Left, cartoon showing the organization of the schistosome tegument. Right, fluorescence in situ hybridization and DAPI labeling overlaid on a Differential Interference Contrast (DIC) micrograph showing the distribution of *tsp-2*+ cells relative to the tegument. Although some cells expressing lower levels of *tsp-2* are located more internally, a majority of *tsp-2*+ cells were located just beneath the parasite muscle layer. (c) Double fluorescence in situ hybridization showing co-localization of *tsp-2* with the indicated tegumental factors. Images

*Figure 2 continued on next page*

*Figure 2 continued*
are representative of parasites (n > 3) recovered from two separate groups of mice. Scale bars: (a) 100 µm, (b, c) 20 µm.
The following figure supplement is available for figure 2:

**Figure supplement 1.** DIS genes are expressed in a population of cells that is distinct from the neoblasts.

nucleated cell bodies that sit in the mesenchyme, beneath the parasite's body-wall muscles (*Morris and Threadgold, 1968*; *Wilson and Barnes, 1974*) (*Figure 2b*). To determine if these DIS genes are expressed in a tegument-associated cell population, we performed double FISH experiments. We first examined the distribution of the mRNA for a Tetraspanin, TSP-2, that encodes a well-characterized tegument-specific factor (*Braschi and Wilson, 2006*; *Tran et al., 2006*; *Pearson et al., 2012*). TSP-2 is currently being explored as an anti-schistosome vaccine candidate due to its presence on the parasite surface (*Hotez et al., 2010*). Consistent with *tsp-2* being expressed in a tegument-associated cell population, we found that a majority of *tsp-2*$^+$ cells are located immediately beneath the parasite's body-wall muscle layer (*Figure 2b*). To further examine this *tsp-2*$^+$ cell population, we performed double FISH with other DIS genes known to encode proteins expressed in the tegument. We observed that DIS genes encoding a panel of known tegumental factors, including *sm13* (Smp_195190), *sm29* (Smp_072190), *sm25* (Smp_195180), an amino acid transporter (Smp_176940) (*Wilson, 2012*), a dysferlin protein (Smp_141010) (*Braschi and Wilson, 2006*; *Wilson, 2012*), an endophillin B1 (*Castro-Borges et al., 2011*; *Wilson, 2012*), and a cd59-like molecule (Smp_081920) (*Wilson, 2012*) were expressed in a largely overlapping population of cells with *tsp-2* immediately beneath the dorsal body-wall muscles (*Figure 2c*). Given their position in the parasite, and their expression of many known tegumental genes, our data indicate that *tsp-2*$^+$ cells represent a population of tegument-associated cells.

Our data suggest that *tsp-2*$^+$ cells co-express many known tegumental factors and are lost within a few days following stem cell depletion. We envision two models to explain these observations. First, *tsp-2*$^+$ cells could represent a relatively long-lived population that requires the continual presence of the somatic neoblasts for their survival. Alternatively, the *tsp-2*$^+$ cells could be a short-lived cell population that requires a pool of stem cells for its continuous renewal. In the absence of this renewal, the *tsp-2*$^+$ cells are rapidly depleted. To distinguish between these possibilities, we performed pulse-chase experiments with the thymidine analogue EdU (*Salic and Mitchison, 2008*). This approach allows us to specifically label neoblasts at S-phase and monitor their differentiation over time (*Collins et al., 2013*). In these experiments, parasite-infected mice were injected with EdU and the distribution of EdU$^+$ cells relative to the *tsp-2*$^+$ cells was monitored every other day for 11 days (*Figure 3a*). If the *tsp-2*$^+$cells are long-lived and turn over slowly, we would anticipate that few *tsp-2*$^+$cells would become EdU$^+$ over the chase period. However, if these cells were renewed rapidly by the neoblasts, we would expect a large fraction of *tsp-2*$^+$ cells to become EdU$^+$. Furthermore, since EdU levels are diluted following cell division, over time differentiating neoblasts would contain less EdU, resulting in a reduction in the EdU levels in *tsp-2*$^+$ cells.

At D1 following an EdU pulse, <0.25% of *tsp-2*$^+$ cells were EdU$^+$ (*Figure 3b,c*), indicating that the *tsp-2*$^+$ cells are not proliferative. After a 3-day chase period, however, we noted that over 40% of *tsp-2*$^+$cells were newly born EdU$^+$ cells (*Figure 3b,c*). This result suggests that stem cells initially incorporating EdU were capable of replenishing nearly half of the *tsp-2*$^+$ cells within three days. Beyond D5, we noted a rapid reduction in the number of *tsp-2*$^+$ EdU$^+$ cells and an overall reduction in the EdU levels per cell (*Figure 3b,c*). These data suggest that *tsp-2*$^+$ cells are a short-lived cell population that is continuously renewed by the neoblasts during the parasite's life in its definitive host. To determine if this rapid rate of neoblast-driven renewal was unique to *tsp-2*$^+$ cells, we also examined the kinetics of EdU labeling of the schistosome intestine. In contrast to the *tsp-2*$^+$ cells, only 2.5% of *cathespin B*$^+$ intestinal cells were EdU$^+$ at D3, and this level remained fairly constant throughout the 11D time course (*Figure 3b,d*). Thus, the kinetics of tegumental cell birth differs from that of intestinal cells. Taken together, our data suggest that on a population level neoblasts are 'biased' toward the rejuvenation of *tsp-2*$^+$ cells over other lineages.

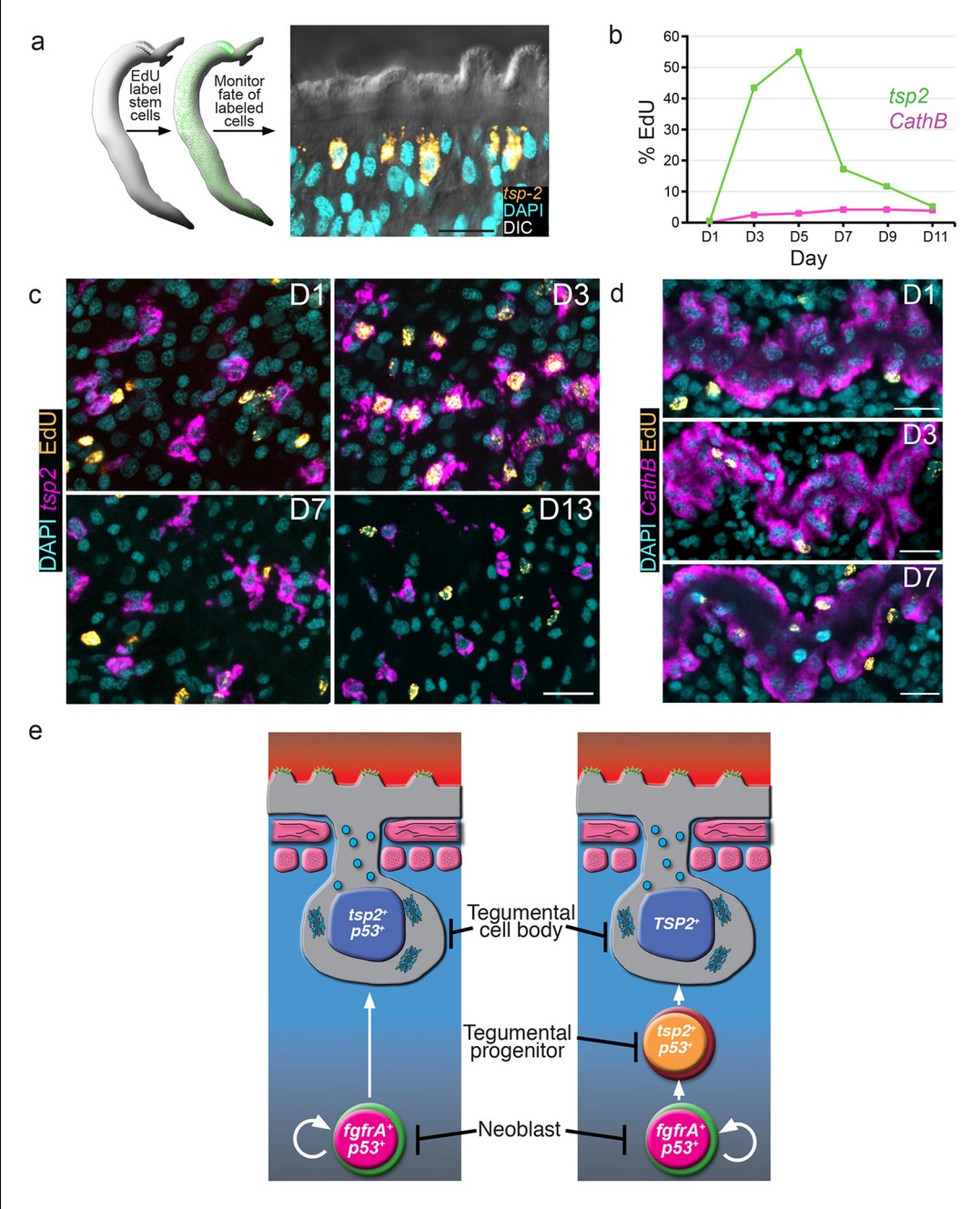

**Figure 3.** *tsp-2*[+] cells are renewed by stem cells and then rapidly turned over. (**a**) Cartoon showing EdU pulse-chase strategy to examine the differentiation of stem cells into *tsp-2*[+] cells. (**b**) Quantification of the number of EdU[+]*tsp-2*[+]or EdU[+]*cathepsin B*[+] cells following a single pulse of EdU given to parasites in vivo. Percentages of EdU[+] *tsp-2*[+]/total *tsp-2*[+] cells were D1 0.22% (2/917), D3 41% (323/787), D5 52% (299/575), 13% (57/437), D9 8.1% (49/603), D11 1.4% (8/567). Percentages of EdU[+] *Cathepsin B*/ total *Cathepsin B*[+] cells were D1 0% (0/1570), D3 2.4% (26/1057), D5 2.9% (61/2044), D7 4.2% (58/1359), D9 4.3% (106/2469), D11 3.9% (64/1646). Data were collected from ≥ 5 male parasites recovered from two separate mice, except for *cathepsin B* labeling at D11 where parasites were recovered from a single mouse. (**c, d**) Fluorescence in situ hybridization showing the EdU labeling of *tsp-2*[+] or *cathepsin B*[+] cells at various time points following an EdU pulse. Scale bars, 15 µm. (**e**) Potential models for tegumental cell differentiation.

In planarians, a population of postmitotic neoblast progeny displays similar sensitivity to irradiation as this tsp-2$^+$ tegument-associated cell population (*Eisenhoffer et al., 2008*). These planarian neoblast progeny similarly express a p53-like protein as well as a large collection of planarian-specific molecules (*Eisenhoffer et al., 2008*; *Zhu et al., 2015*). Most importantly, these planarian cells serve as progenitors to terminally differentiated epidermal cells (*van Wolfswinkel et al., 2014*; *Tu et al., 2015*). Thus, it appears that free-living and parasitic flatworms utilize similar developmental strategies for epidermal maintenance. Presently, electron microscopy is the only methodology to unambiguously identify tegumental cell bodies in schistosomes. Therefore, with current technology, it is not possible to determine which of the irradiation-sensitive cells expressing DIS genes are terminally differentiated tegumental cell bodies. In light of this limitation, our data are consistent with two models (*Figure 3e*). In the first model, proliferating neoblasts differentiate to produce a short-lived population of terminally differentiated tegumental cell bodies expressing tsp-2 and other DIS genes (i.e., sm13, sm25, sm29, etc.). In the alternative model, cells expressing tsp-2 (and other DIS genes) represent a population of short-lived progenitors to terminally differentiated tegumental cell bodies. Regardless of which model (or combination of these models) is correct, our data suggest that a primary function of the schistosome neoblasts is to generate cells that contribute to the tegument.

The mammalian bloodstream would appear to be a rather inhospitable niche for a pathogen. In the case of schistosomes, there is little dispute about the importance of the tegument in defending the parasite from host immunity (*McLaren, 1980*; *Skelly and Alan Wilson, 2006*), yet the properties of this tissue that afford the parasite protection in blood are unclear. Indeed, numerous ideas have been proposed to explain this phenomenon, including tegumental absorption of host antigens (*Smithers et al., 1969*; *Clegg et al., 1971*) and the turnover of the unique tegumental outer membranes (*Perez and Terry, 1973*; *Wilson and Barnes, 1977*). Based on our data, we suggest that neoblast-driven tegumental regeneration may play a key role in the parasite's ability to survive in the mammalian host. By undergoing continuous tegumental renewal, the parasite is likely capable of rapidly turning over the tegument and regenerating damage inflicted inside the host (e.g., via immune attack). Thus, an important goal for future studies is to address the role of neoblasts in parasite survival and tegumental function in vivo.

## Materials and methods

### Parasite acquisition and culture

Adult *S. mansoni* (6–8 weeks post-infection) were obtained from infected mice by hepatic portal vein perfusion with 37°C DMEM (Mediatech, Manassas, VA) plus 5% Fetal Bovine Serum (FBS, Hyclone/Thermo Scientific Logan, UT) and heparin (200–350 U/ml). Parasites were rinsed several times in DMEM + 5% FBS and cultured (37°C/5% CO$_2$) in Basch's Medium 169 (*Basch, 1981*) and 1x Antibiotic-Antimycotic (Gibco/Life Technologies, Carlsbad, CA 92008). Media were changed every 1–3 days.

### γ-irradiation, RNAi and transcriptional profiling

For transcriptional profiling of irradiated worms, parasites were harvested from mice, suspended in Basch medium 169, and exposed to 200 Gy of γ-irradiation using a Gammacell-220 Excel with a Co$^{60}$ source (Nordion, Ottawa, ON, Canada). Control parasites were mock irradiated. Parasites were cultured in Basch Medium 169 and 48 hr or 14D post-irradiation males were separated from female parasites using 0.25% ethyl 3-aminobenzoate methanesulfonate. Following separation, the head and testes of males were amputated with a sharpened tungsten needle (*Collins et al., 2013*) and purified total RNA was prepared from the remaining somatic tissue from pools of 9–18 parasites using Trizol (Invitrogen, Carlsbad, CA) and DNase treatment (DNA-free RNA Kit, Zymo Research, Irvine, CA). Three independent biological replicates were performed for both control and irradiated experimental groups. For RNAi of fgfrA and histone H2B, parasites were treated with dsRNA as previously described (*Collins et al., 2013*), and RNA was extracted at D21 using similar procedures as used for the irradiated parasites. Detailed files of the RNAseq results can be found in *Supplementary file 2*. Three biological replicates were performed for fgfrA(RNAi) and two biological replicates for H2B (RNAi). Control RNAi treatments with an irrelevant dsRNA synthesized from the ccdB and camR-

containing insert of plasmid pJC53.2 (*Collins et al., 2013*) were performed alongside *fgfrA* and *H2B* dsRNA treatments.

To measure transcriptional differences, RNAseq analysis was performed on an Illumina HiSeq2000 and analyzed using CLC Genomics Workbench as described previously (*Collins et al., 2013*). To define genes down-regulated in all treatment groups and genes specifically down-regulated following long-term stem cell depletion, we first compared the lists of genes down-regulated (>1.25 fold change, p<0.05, t-test) at D2 and D14 post-irradiation. This list was then cross-referenced to our RNAi datasets to define the DIS genes and the 135 genes down regulated in both the irradiation and RNAi treatments. To reduce false negatives we only required genes to be significantly down-regulated in either the *fgfrA(RNAi)* or the *H2B(RNAi)* treatments. For quantification of gene expression RNA was reverse transcribed (iScript, Biorad) and quantitative real-time PCR was performed on a BioRad CFX96 Real Time System with iTaq Universal SYBR Green Supermix (Biorad). Relative expression was determined using the $\Delta\Delta Ct$ method and mean $\Delta Ct$ values of biological replicates were used to make statistical comparisons between treatments. Oligonucleotide sequences are listed in *Supplementary file 3*.

### Parasite labeling and Imaging

Whole-mount in situ hybridization and EdU labeling of parasites grown in mice were performed as previously described (*Collins et al., 2013*) Tyramide Signal Amplication for double fluorescence in situ hybridization was performed essentially as previously described ( *Collins et al., 2010*) except 100mM sodium azide was used to quench peroxidase activity between rounds of signal development. cDNAs used for RNAi or in situ hybridization were cloned in plasmid pJC53.2 using TA-based cloning (*Collins et al., 2010*) or Gibson assembly (New England Biolabs Gibson Assembly Master Mix, E2611S); oligonucleotide primer sequences are listed in *Supplementary file 3*. Imaging of specimens was performed similar to previous studies (*Collins et al., 2010*; *2011*) using either a Zeiss LSM 710 or Zeiss LSM 700 for confocal imaging or a Leica MZ205 or Zeiss AxioZoom for brightfield imaging. For whole-mount in situ hybridizations on irradiated parasites, parasites were recovered from mice, exposed to 200 Gy of X-ray irradiation using a CellRad irradiator (Faxitron Bioptics, Tucson, AZ) or 100 Gy of Gamma Irradiation on a J.L. Shepard Mark I-30 $Cs^{137}$ source, and cultured in vitro for indicated periods of time.

## Acknowledgements

We thank Melanie Issigonis and Bo Wang for comments on the manuscript, as well as Alvaro Hernandez and the High-Throughput Sequencing Unit of the Roy J Carver Biotechnology Center for Illumina sequencing. This work was supported by NIH R21AI099642 (PAN). Mice and *B. glabrata* snails were provided by the NIAID Schistosomiasis Resource Center of the Biomedical Research Institute (Rockville, MD) through NIH-NIAID Contract HHSN272201000005I for distribution through BEI Resources. PAN is an investigator of the Howard Hughes Medical Institute. In adherence to the Animal Welfare Act and the Public Health Service Policy on Humane Care and Use of Laboratory Animals, all experiments with and care of vertebrate animals were performed in accordance with protocols approved by the Institutional Animal Care and Use Committee (IACUC) of the University of Illinois at Urbana-Champaign (protocol approval number 10035) and UT Southwestern Medical Center (protocol approval number APN 2014-0072). RNAseq datasets are available at NCBI under the accession numbers GSE42757 and GSE75861. The authors declare no competing interests.

## Additional information

### Funding

| Funder | Grant reference number | Author |
| --- | --- | --- |
| Howard Hughes Medical Institute | Investigator | Phillip A Newmark |
| National Institutes of Health | R21AI099642 | Phillip A Newmark |

The funders had no role in study design, data collection and interpretation, or the decision to submit the work for publication.

## Author contributions

JJC, GRW, Conception and design, Acquisition of data, Analysis and interpretation of data, Drafting or revising the article; HI, Acquisition of data, Analysis and interpretation of data, Drafting or revising the article; PAN, Conception and design, Analysis and interpretation of data, Drafting or revising the article

## Ethics

Animal experimentation: In adherence to the Animal Welfare Act and the Public Health Service Policy on Humane Care and Use of Laboratory Animals, all experiments with and care of vertebrate animals were performed in accordance with protocols approved by the Institutional Animal Care and Use Committee (IACUC) of the University of Illinois at Urbana-Champaign (protocol approval number 10035) and UT Southwestern Medical Center (protocol approval number APN 2014-0072).

# Additional files

## Supplementary files

• Supplementary file 1. Spreadsheet detailing genes down-regulated at early and late time points after stem cell depletion.

• Supplementary file 2. Spreadsheet containing detailed expression information from Day 14 post-irradiation, *histone H2B(RNAi),* and *fgfrA(RNAi)* RNAseq experiments.

• Supplementary file 3. Spreadsheet detailing oligonucleotide primers used in this study.

## Major datasets

The following dataset was generated:

| Author(s) | Year | Dataset title | Dataset URL | Database, license, and accessibility information |
|---|---|---|---|---|
| Collins JJ, Newmark PA | 2015 | Stem cells rejuvinate the schistosome host-parasite interface | http://www.ncbi.nlm.nih.gov/geo/query/acc.cgi?acc=GSE75861 | Publicly available at Gene Expression Omnibus (Accession no: GSE75861). |

The following previously published dataset was used:

| Author(s) | Year | Dataset title | Dataset URL | Database, license, and accessibility information |
|---|---|---|---|---|
| Collins JJ, Wang B, Lambrus BG, Tharp ME, Iyer H, Newmark PA | 2013 | Adult stem cells in the human parasite Schistosoma mansoni | http://www.ncbi.nlm.nih.gov/geo/query/acc.cgi?acc=GSE42757 | Publicly available at Gene Expression Omnibus (Accession no: GSE42757). |

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
