## [Decision Letter]

Thank you for submitting your work entitled "Stem cell progeny contribute to the schistosome host-parasite interface" for consideration by *eLife*. Your article has been reviewed by two peer reviewers, including Alex Loukas, and the evaluation has been overseen by Alejandro Sánchez Alvarado as the Reviewing Editor and Janet Rossant as the Senior Editor.

The reviewers have discussed the reviews with one another and the Reviewing Editor has drafted this decision to help you prepare a revised submission.

Summary:

The present manuscript by Collins et al., extends on an earlier discovery by the authors on the presence of adult somatic stem cells in schistosome blood flukes. Here they report on the participation of these cells in the constant renewal of the tegument of these organisms. The authors show that the adult stem cells (neoblasts) generate a short-lived cell population of terminally differentiated tegumental cell bodies/progenitors expressing several of formerly characterized tegumental factors associated with the parasite's surface. The study sheds new light on the function of neoblasts for parasite survival. This is a significant contribution to our understanding of how these organisms can survive for decades within the hostile bloodstream of its final mammalian host while defending immune attacks. Moreover, given the importance of many of the molecules found in the tegument in terms of vaccine discovery and development, the work presented here on their biogenesis has important ramifications for the development of future control strategies for this important human disease.

There are, however, a few essential revisions the authors need to address in order for this well-designed study to be ultimately published in *eLife*.

Essential revisions:

1) There is some confusion regarding gene annotations among Figure 1, [Supplementary-material SD1-data], and [Supplementary-material SD2-data]:

In contrast to *cyclinB* and *mcm2, nanos2* and *ago2-1* cannot be found in [Supplementary-material SD1-data] (please comment);

Smp_179320 (eukaryotic translation initiation factor 2C, [Supplementary-material SD1-data]) is annotated as *ago2* in GeneDB;

Smp_051920 (hypothetical protein, [Supplementary-material SD1-data]) is annotated as *nanos1* (not *nanos2*) in GeneDB;

Smp_157300 (tyrosine kinase, [Supplementary-material SD1-data]) is annotated as basic fibroblast growth factor receptor 1 A in GeneDB; is this corresponding to *fgfrA* targeted by RNAi?

Why can *histone H2B* not be found in [Supplementary-material SD1-data] although it was a target of RNAi?;

Smp_181530 (tetraspanin, [Supplementary-material SD2-data]) is annotated as *cd63* antigen (tetraspanin family member) in GeneDB; does this gene correspond to *tsp-2* (Figure 1)?

Smp_141010 is annotated as *dysferlin1* in GeneDB whereas it is named *fer-1-*related in [Supplementary-material SD2-data].

2) Many genes are downregulated by *h2b(RNAi)* and *fgfrA(RNAi)* to a comparable extent, e.g. Smp_133990 (topbb1, [Supplementary-material SD1-data]) and Smp_174820 (Ser/Thr kinase, [Supplementary-material SD1-data]). These genes show also a similar downregulation after 48h and D14 post-irradiation. However, some genes, e.g. Smp_095290 (zinc finger protein, [Supplementary-material SD1-data]), Smp_194180 (hypothetical protein, [Supplementary-material SD1-data]), and Smp_139530 (cellular tumor antigen P53, [Supplementary-material SD1-data]) show a much higher transcriptional downregulation following *h2b(RNAi)* in comparison to *fgfrA(RNAi).* How can this be explained? Is this due to differences of RNAi efficacies (have these been determined by real time PCR)?

3) Interestingly, this finding is paralleled by a significantly higher downregulation after D14 in comparison to 48h post-irradiation. How can this be explained (e.g. by mRNA stability?), and is this somehow related to the former observation (differences in RNAi efficacies)?

4) How can the differing reductions of *histone H2A* transcription levels (Smp_086860 and Smp_002930) be explained ([Supplementary-material SD1-data])?

The observation of deviating transcriptional downregulation following *h2b(RNAi)* in comparison to *fgfrA(RNAi)* can also be seen for genes only down-regulated at late time points ([Supplementary-material SD2-data]), e.g. SmVAL8, Sm25, Sm13, and SmVAL12 (Smp_123550, Smp_195180, Smp_195190, and Smp_123540). Moreover, SmVAL8 transcription seems to be slightly but significantly increased after 48h post-irradiation before drastically downregulated (D14). How can this be explained?

5) One would expect that the ablation of neoblasts and the following decreased replenishment of tegumental cell bodies impede surface renewal. As a consequence this would lead to morphological changes at the surface area. Has this been observed microscopically?

---

## [Author Response]

*1) There is some confusion regarding gene annotations among Figure 1, [Supplementary-material SD1-data], and [Supplementary-material SD2-data]: In contrast to cyclinB and mcm2, nanos2 and ago2-1 cannot be found in [Supplementary-material SD1-data] (please comment); Smp_179320 (eukaryotic translation initiation factor 2C, [Supplementary-material SD1-data]) is annotated as ago2 in GeneDB; Smp_051920 (hypothetical protein, [Supplementary-material SD1-data]) is annotated as nanos1 (not nanos2) in GeneDB; Smp_157300 (tyrosine kinase, [Supplementary-material SD1-data]) is annotated as basic fibroblast growth factor receptor 1 A in GeneDB; is this corresponding to fgfrA targeted by RNAi?; Smp_181530 (tetraspanin, [Supplementary-material SD2-data]) is annotated as cd63 antigen (tetraspanin family member) in GeneDB; does this gene correspond to tsp-2 (Figure 1)? Smp_141010 is annotated as dysferlin1 in GeneDB whereas it is named fer-1-related in [Supplementary-material SD2-data].*

We have updated the annotations in [Supplementary-material SD1-data] and [Supplementary-material SD2-data] to include the most recent annotations available from the curators at the Wellcome Trust Sanger Institute. In cases where we have ascribed names to genes (e.g. *fgfrA* and *nanos2*) we have contacted the genome curators and the annotations in GeneDB have been updated.

*Why can histone H2B not be found in [Supplementary-material SD1-data] although it was a target of RNAi?;* To address this point we’ve included the entire H2B RNAi dataset (in addition to the *fgfrA (RNAi)* and D14 irradiation datasets) as a new supplemental table ([Supplementary-material SD3-data]). In this dataset the *h2b* mRNA (Smp_108390) is down regulated > 15 fold, verifying that our knockdown was successful. Why is *h2b* not significantly down regulated in our other datasets? We attribute this observation to basal levels of histone expression in non-neoblast cells. Although neoblasts express high levels of *histone h2b*, as a population they only represent a small fraction (~5%) of cells in the animal. Thus, neoblast ablation (by for example *fgfrA* RNAi or irradiation) fails to completely reduce the *H2B* mRNA levels while still drastically reducing the number of neoblasts expressing high levels of *H2B* transcript (See Collins et al., 2013 Nature 494, 476–479, Figure 4C vs. Figure S8).

*2) Many genes are downregulated by h2b(RNAi) and fgfrA(RNAi) to a comparable extent, e.g. Smp_133990 (topbb1, [Supplementary-material SD1-data]) and Smp_174820 (Ser/Thr kinase, [Supplementary-material SD1-data]). These genes show also a similar downregulation after 48h and D14 post-irradiation. However, some genes, e.g. Smp_095290 (zinc finger protein, [Supplementary-material SD1-data]), Smp_194180 (hypothetical protein, [Supplementary-material SD1-data]), and Smp_139530 (cellular tumor antigen P53, [Supplementary-material SD1-data]) show a much higher transcriptional downregulation following h2b(RNAi) in comparison to fgfrA(RNAi). How can this be explained? Is this due to differences of RNAi efficacies (have these been determined by real time PCR)?*

In our experience RNAi of *H2B* rapidly results in a complete loss of proliferative neoblasts whereas RNAi of *fgfrA* fails to ablate the entire neoblast pool (see Collins et al., (2013) Nature 494, 476–479). Therefore, we fully expect that *h2b* RNAi will result in a larger fold change down regulation for neoblast expressed and DIS genes even though both *H2B* and *fgfrA* genes are down-regulated >10 fold in our RNAseq datasets ([Supplementary-material SD3-data]).

*3) Interestingly, this finding is paralleled by a significantly higher downregulation after D14 in comparison to 48h post-irradiation. How can this be explained (e.g. by mRNA stability?), and is this somehow related to the former observation (differences in RNAi efficacies)?*

The reviewers are correct, this is a very interesting observation and we have added additional data analyzing Smp_139530 (*p53*) to the manuscript (Figure 2 and Figure 2—figure supplement 1). We find that in addition to being expressed in neoblasts the schistosome *p53* homolog is also highly expressed in *tsp-2* cells. Based on this observation we interpret our transcriptional profiling results as follows: At 48 hours following irradiation *p53* is partially down-regulated due to neoblast loss and then by day 14 the transcript is highly down regulated due to loss of the *tsp-2^+^* cells. This result also adds value to the paper since the planarian orthologue of this gene is expressed in a similar manner (i.e., in the neoblasts and their immediate progeny, see Pearson & Sánchez Alvarado (2010) Development 137, 213–221). This provides additional evidence supporting the evolutionary conservation of the mechanisms regulating planarian and schistosome epidermal lineages.

*4) How can the differing reductions of histone h2a transcription levels (Smp_086860 and Smp_002930) be explained ([Supplementary-material SD1-data])?*

We have examined the sequences of Smp_086860 and Smp_002930 and they appear to encode two distinct Histone H2A genes with differing amino acid sequences. Why do we observe differing levels of fold change down regulation? One possibility is that one of these genes is a histone variant. In other organisms histone variants can be regulated differently than the conical histones (see Kamakaka and Biggins. (2005) Genes and Dev. 19:295-316). Indeed, Smp_002930 shares >90 identity to histone H2A isoforms in Humans, whereas Smp_086860 only shares about 75-80% identity with human H2A proteins. Thus, Smp_086860 may encode a neoblast-enriched H2A variant.

*The observation of deviating transcriptional downregulation following h2b(RNAi) in comparison to fgfrA(RNAi) can also be seen for genes only down-regulated at late time points ([Supplementary-material SD2-data]), e.g. SmVAL8, Sm25, Sm13, and SmVAL12 (Smp_123550, Smp_195180, Smp_195190, and Smp_123540).*

This comment was addressed in point 2.

*Moreover, SmVAL8 transcription seems to be slightly but significantly increased after 48h post-irradiation before drastically downregulated (D14). How can this be explained?*

We have examined this observation in more detail by performing qPCR at 48 hours post-irradiation (Figure 1—figure supplement 2) and in situhybridizations for *val-8* at both 48 hours and 7 days post-irradiation (Figure 2). We observe that *val-8* is expressed at slightly higher levels at 48 hours post-irradiation by qPCR yet doesn’t seem to be expressed in noticeably more cells at the same time point by in situ hybridization. Thus, we speculate that *val-8* mRNA maybe present at slightly higher levels in the *val-8^+^* cells at 48 hours. Based on this we speculate that irradiation may alter the differentiating neoblasts such that they turn on some differentiation markers (e.g., *val-8*) at slightly higher levels.

5) One would expect that the ablation of neoblasts and the following decreased replenishment of tegumental cell bodies impede surface renewal. As a consequence this would lead to morphological changes at the surface area. Has this been observed microscopically?

This is an excellent point and a major priority of our ongoing efforts. However, there are several challenges we need to tackle before we can address this issue in a comprehensive manner and therefore we feel this extends beyond the scope of this manuscript. We first must determine when the tegumental cell bodies are depleted following neoblast loss. The data presented here do not address this point. In principle, the *tsp-2* mRNA^+^ cells could represent the entire tegumental cell pool, a subset of the tegumental cell bodies, or progenitors to the definitive tegument (Figure 3). Addressing this point will allow us to determine the proper time point to examine the tegument microscopically. As far as microscopic analyses the preferred methodology would be Transmission Electron Microscopy (TEM), ideally using the methodology developed by Hockley and McLaren (Hockley DJ, McLaren DJ. 1973 Int J Parasitol. 3(1):13-25) that allows for visualization of both the inner and outer tegumental membranes. Unfortunately, our efforts to robustly reproduce this approach have yet to be successful, and therefore we are exploring other methodologies, including High Pressure Freezing and Freeze Substitution that should better preserve the membranes. Finally, since it has been noted that the tegument does not behave normally in vitro (see pg., 252 in Skelly, P. J. and R. A. Wilson (2006). "Making sense of the schistosome surface." Adv Parasitol 63: 185-284) the ideal experiment would be to examine the tegument of neoblast-depleted parasites during the course of an experimental infection. While we are developing the methodologies to perform these experiments, they are still at least a year from reaching a publishable endpoint.